# Loss of βENaC Prevents Hepatic Steatosis but Promotes Abdominal Fat Deposition Associated with a High-Fat Diet

**DOI:** 10.3390/biology14111558

**Published:** 2025-11-06

**Authors:** Madison Hamby, Elizabeth Barr, Seth Lirette, Heather A. Drummond

**Affiliations:** 1Department of Physiology and Biophysics, University of Mississippi Medical Center, Jackson, MS 39216-4505, USA; mnewberry@umc.edu (M.H.); mebarr@umc.edu (E.B.); 2Department of Data Sciences, University of Mississippi Medical Center, Jackson, MS 39216-4505, USA; slirette2@umc.edu

**Keywords:** degenerin, hepatic steatosis, metabolic homeostasis, energy balance

## Abstract

**Simple Summary:**

The purpose of this study was to determine if loss of βENaC impacts energy balance and fat distribution associated with a high-fat diet. We examined body fat and lean mass, energy expenditure, food consumption, motor activity, and fat distribution in the liver and surrounding the gonads and abdominal organs. We found that male mice with reduced expression of βENaC tended to weigh less and have less fat deposition in the liver than controls. However, male mice with reduced βENaC had larger gonadal and abdominal organ fat depots. These findings suggest that βENaC plays a novel role in fat distribution, an important determinant of metabolic health.

**Abstract:**

Background: Degenerin proteins, such as Acid-Sensing Ion Channel 2 (ASIC2) and β Epithelial Na^+^ Channel (βENaC), have been implicated in cardiovascular function. We previously demonstrated that mice lacking normal levels of βENaC and ASIC2 are protected from diet-induced obesity, metabolic disruption, and hepatic steatosis. Methods: To investigate the specific role of βENaC proteins in the progression of metabolic disease, we examined the impact of a high-fat diet (HFD) in the βENaC hypomorph mouse model (βMUT). Body composition and metabolic and behavioral phenotypes were examined in male and female and βMUT and WT mice (n = 6–14/group) fed a normal chow diet (NFD) from weaning until 16 weeks of age, then a 60% kcal-fat diet for 5 weeks. Results: Compared to WT mice, βMUT male mice have reduced lean and total body mass. No remarkable differences in energy expenditure, motor activity, or food consumption patterns were detected. HFD-fed male βMUT mice exhibited reduced liver fat content (mass and Oil Red O staining) yet increased abdominal fat depots. HFD-fed female βMUT mice exhibited lower heart mass. Conclusions: These novel findings suggest a role for βENaC in the maintenance of metabolic homeostasis and adipose tissue distribution.

## 1. Introduction

Metabolic syndrome consists of a group of risk factors that increase the likelihood of developing obesity, heart disease, type 2 diabetes, chronic kidney disease, liver disease, and stroke [1,2]. With the prevalence of metabolic syndrome continuing to rise worldwide, it is important to understand the underlying mechanisms of this disease [3].

Degenerin proteins form sodium and cation channels, and they are expressed in diverse cell types that participate in cardiometabolic homeostasis, including immune cells, smooth muscle cells, endothelial cells, epithelial cells, and neurons, and contribute to diverse features of cardiovascular control, including water and electrolyte homeostasis, vascular tone, and autonomic control [4,5,6,7,8,9,10,11,12]. Our lab has demonstrated that mice lacking normal levels of two degenerin ion channels, Acid-Sensing Ion Channel 2 (Asic2) and beta Epithelial Sodium Channel (βENaC), were protected from weight gain, loss of insulin sensitivity, and liver steatosis during high-fat diets [13]. The contribution of individual degenerin proteins to this metabolic phenotype is unclear. In a recently published investigation, we determined that Asic2-knockout mice show modest protection from high-fat diet-induced weight gain, but no protection from hepatic steatosis [14]. Thus, the purpose of this investigation was to determine the contribution of βENaC to protection from diet-induced obesity and hepatic steatosis.

The hypothalamus plays a central role in energy balance. Several hypothalamic regions coordinate energy homeostasis including the arcuate nucleus, ventromedial hypothalamus, paraventricular nucleus, and lateral hypothalamic area. Using RNA-seq data from HypoMap, a publicly available database, we determined that Asic2 is the predominant degenerin protein expressed in the hypothalamus and is expressed in 30–60% of metabolically active arcuate neurons, including those expressing receptors for the hunger-signaling hormone ghrelin and the satiety-signaling hormones leptin and insulin [15]. The localization of βENaC in these neuron populations has never been addressed. Thus, one aim of this study was to determine βENaC localization in the mouse hypothalamus using the HypoMap dataset. The second aim of this study was to determine if βENaC contributes to energy and metabolic homeostasis, diet-induced obesity, and hepatic steatosis in a hypomorph βENaC mouse model. Our findings suggest a minimal role for βENaC in energy balance and the development of diet-induced obesity; however, reduction of βENaC expression prevented hepatic steatosis, suggesting a potential role for βENaC in the progression of metabolic dysfunction-associated fatty liver disease.

## 2. Materials and Methods

Localization of hypothalamic βENaC mRNA (Scnn1b). We used the HypoMap dataset, an integrated hypothalamic reference map of 384,925 cells, to identify hypothalamic neuron regions expressing βENaC [15]. The dataset and an explorer tool are available at https://cellxgene.cziscience.com/collections (accessed on 24 June 2025). The RDS file was downloaded from https://www.repository.cam.ac.uk/items/8f9c3683-29fd-44f3-aad5-7acf5e963a75 (accessed on 10 August 2024) and is available from the Apollo Repository hosted by the University of Cambridge. The RDS file was analyzed using Seurat 5 in R Studio (Version 2024.04.2+764). We generated dimensional reduction, or Uniform Manifold Approximation and Projection (UMAP), clusters that are labeled for cell type based on the “C7_named” clustering level in the meta data [15]. We generated feature and dot plots to identify hypothalamic regions, and a violin plot to identify the cell types expressing Scnn1b.

Experimental animals. All animal work was conducted under the Association for Assessment and Accreditation of Laboratory Animal Care (AAALAC) and the University of Mississippi Medical Center (UMMC) Institutional Animal Care and Use Committee (IACUC). The βENaC hypomorph model (βMUT, provided by Dr. Hummler, University of Lusanne, Switzerland) [16] has been maintained in our laboratory since 2008 [8,13,14,17,18,19,20,21]. Animals were maintained as homozygous mating pairs. Mice were kept under a 12 h light/12 h dark–light cycle and permitted normal chow and water ad libitum. Male and female mice were utilized for all protocols. All protocols were approved by the Institutional Animal Care and Use Committee at UMMC.

The timeline for our experimental approach is shown in Figure 1. Mice were raised on normal chow (14% kcal fat, 54% kcal carbohydrate, Envigo, Toronto, Ontario, Canada, Cat #TD.8604), then body composition (EchoMRI, 4-in-1 EchoMRI-900TM, Echo Medical System, Houston, TX, USA) was determined at 16 weeks of age. EchoMRI uses nuclear magnetic resonance to assess total body fat and water content. Conscious mice are weighed, then placed in a clear tube to limit movement and the scan initiated. The average of two measurements was used. Body composition was determined under fed conditions at the same time each day. Mice were placed in metabolic cages for assessment of indirect calorimetry the following day. Mice were then fed a 5-week high-fat diet (HFD; 60% kcal from fat, 21% kcal from carbohydrate, Envigo, Cat #TD.06414) and body composition and indirect calorimetry were reassessed at 21 weeks of age. Within 3–5 days, a glucose tolerance test was conducted, then organs and plasma samples were collected.

Indirect calorimetry, activity, and feeding measurements. Mice were placed in metabolic cages (Sable Systems Promethion Core, Sable Systems, North Las Vegas, NV, USA) following their body composition measurements to assess metabolic and behavioral phenotypic changes in response to normal chow and again following the HFD. Mice were housed individually in temperature-controlled metabolic cages with ad libitum access to food and water and given a 48 h acclimation period before 72 h of continuous recording. The averages of the three 24 h recorded periods were used as values for each animal. Individual cages contained a ceiling-mounted food hopper, water spigot, and “hut” that weighed the mice. Food and water were loaded at an amount great enough to allow cages to remain closed for the length of the study. This system allows for high-resolution measurements of oxygen consumption and carbon dioxide production. Energy expenditure (kcal/day) was calculated from VO_2_ and VCO_2_ data using the Weir equation: 3.941 × VO_2_ (L/day) + 1.106 × VCO_2_ (L/day). To estimate basal metabolic rate (BMR), we used the mean energy expenditure during the lowest energy expenditure 30 min period (R_EE_30 in Promethion Data Analysis Guide). RER is defined as the mean respiratory exchange ratio (RER) of VCO_2_ to VO_2_, and this value is unitless. Sable Systems International considers the animal to be “sleeping” when the animal has been “quiet” for more than 40 s, meaning a time in which the animal is not engaged in eating, drinking, grooming, or locomotion [22]. Motor activity was determined using movements measured in the x, y, and z planes, including grooming and scratching. Total food consumption is defined as the average mass of food removed from the food bin over three 24 h periods. One female WT animal was excluded from the results due to failure to gain mass on an HFD. Food consumption data were excluded from the study (n = 2 βMUT females at the NFD timepoint; n = 1 WT female, n = 3 βMUT females, and n = 1 βMUT male at the HFD timepoint) if the values were indicative of a technical malfunction during the acquisition. Post-analysis genotypes were confirmed by PCR using liver DNA. All parameters shown are presented as 24 h averages.

Glucose tolerance test. Following the HFD, a glucose tolerance test was performed on a subset of animals before entering the metabolic cages to assess changes in insulin sensitivity. Mice were fasted for 4 h prior to assessment, injected with 1 g glucose/kg body mass intra-peritoneally, and blood samples, via tail nick, were collected at 0, 15, 30, 60, 90, and 120 min after injections. Blood glucose was assessed using an Accu-check glucose meter, Acon Laboratories, San Diego, CA, USA.

Anthropometric measurements and plasma assays. Upon completion of the study, mice were fasted for 4 h, euthanized with isoflurane until respiration ceased, then the chest was opened and plasma obtained transcardially. The heart, including the atrial appendage, was removed and excess blood blotted and weighed, then the kidneys, liver, spleen, pancreas, and gonadal, visceral, and brown fat pads were collected, weighed, then snap-frozen in liquid nitrogen and stored at −70 °C, except livers. A portion of the liver was fixed in 4% paraformaldehyde for later analysis. To obtain body length, we measured the distance between the tip of the nose to the base of the tail using a mm ruler. To obtain tibial length, we measured the distance from the ankle to the knee using a mm ruler. Plasma leptin (R&D Systems, Cat #MOB00B; RRID:AB_2943468), insulin (Crystal Chem, Elk Grove Village, IL, USA, Cat #90080; RRID:AB_2783626), and ghrelin (Millipore/Sigma, Burlington, MA, USA, Cat #EZRGRT-91K) were assessed by ELISA through the Analytical and Assay Core in the Department of Physiology and Biophysics at UMMC. Plasma cholesterol, low density lipoprotein cholesterol (LDL-C), high density lipoprotein cholesterol (HDL-C), triglycerides, and liver enzymes alanine aminotransferase (ALT), aspartate aminotransferase (AST), and lactate dehydrogenase (LD) were assayed using the Vet Axel Chemistry Analyzer, Alfa Wasserman, West Caldwell, NJ, USA, through the Analytical and Assay Core. The Analytical and Assay Core uses a standardized approach and internal standards to provide a repeatable and reliable assessment.

Liver steatosis measurements. Multiple methods were used to assess liver fat content. Following tissue harvest, livers were immediately assessed for fat content using EchoMRI, then livers were fractioned for −20 °C storage or fixation in 4% paraformaldehyde. Liver triglyceride content was assessed from 100 mg liver samples (AbCam, Waltham, MA, USA, Cat #65336). A portion of the liver was fractioned for fixation for Oil Red O (Sigma Cat #01391) staining in cryosections through the Histology Core in the Department of Physiology and Biophysics at UMMC and quantified as the % field of view area containing a signal using a threshold function in MetaMorph version 5.0 [23,24]. The same threshold was applied to all images. Two fields of view chosen at random were analyzed per mouse, and the average of the two % field of view areas containing signals were determined for each mouse and represented on the graph.

Quantification and statistical analysis. Data were preliminarily analyzed in Excel. Body composition and liver adiposity data were exported to GraphPad Prism version 10.0 for statistical analysis. Preliminary analysis determined that data were normally distributed. Normality was assessed by the Shapiro–Wilk test (*p* > 0.05). Based on this, parametric statistical tests were performed. Due to differences in body mass among groups, covariate adjusted analyses were conducted in STATA version 19.5 for metabolic–behavioral, plasma hormone, and morphometric data using generalized linear models with appropriate families (Gaussian for normal data and gamma for right-skewed data) with fully interacted genotype/sex/diet variables. Marginal means and their associated effects were then estimated from these models. Because all estimates come from a single model and are estimated from pooled post-stratification data, the type-I error rate is controlled and no further post-hoc adjustments are necessary. Modified means, *p* values, and effect sizes for the covariates with the largest effect sizes are provided in table format in respective figures with graphical presentation of absolute values. The absolute data and adjusted means do not align for all measurements. Thus, the covariate analyses were used to discuss outcomes. Liver fat content and plasma hormone data were analyzed using two-way ANOVAs followed by a Fishers LSD Post Hoc Analysis. Comparisons were limited a priori to those within sexes. *p* values for differences between genotypes where *p* < 0.200 are provided in figure panels. Data represent mean ± standard error of the mean. Figures and/or figure legends identify sample sizes, specific statistical and post hoc analyses, and *p* values.

## 3. Results

### 3.1. βENaC Is Expressed in the Murine Hypothalamus

Using the online explorer tool available at CZ CELLxGENE (https://cellxgene.cziscience.com/e/dbb4e1ed-d820-4e83-981f-88ef7eb55a35.cxg/ (accessed on 24 June 2025)) and Seurat in R Studio, we analyzed βENaC mRNA expression in the hypothalamus using RNA-seq data from the HypoMap dataset [15]. As shown in the dimensional reduction plot, this dataset identifies 15 distinct neuron regions and one non-neuronal population within the hypothalamus (Figure 2A), colorized by “Region summarized” categories. Cells expressing βENaC within these populations are identified in red (Figure 2B). The expression rate of βENaC in the murine hypothalamus is low, with only 0.07% of neurons expressing βENaC mRNA. Regions with the greatest percentage of βENaC-expressing neurons included the dorsomedial nucleus > tuberal nucleus > and ventromedial hypothalamic nucleus (Figure 2C). βENaC expression is present in glutamatergic > gabaergic neurons, and to a lesser extent in populations of astrocytes and oligodendrocytes (Figure 2D).

### 3.2. Male βMUT Mice Weigh Less than WT Controls

While βMUT and WT female mice share similar body, fat-free, and fat masses on normal chow and 5-week HFD (Figure 3A,B,D), male βMUT male mice weigh less than WT mice on either diet. The lower body mass is driven by a reduction in fat-free mass (Figure 3B). Body composition was not different between genotypes on either diet (Figure 3C,E).

### 3.3. Metabolic Indices on Normal Chow and HFD Were Mostly Similar Between βMUT and WT Controls

Absolute values for total energy expenditure (kcal/day), RER, basal metabolic rate (kcal/h), motor activity (meters), and sleep time (h) were similar between βMUT and WT mice before and after HFD and are shown in Figure 4A–E. Adjusted for body and fat-free masses, 24 h total energy expenditure, basal metabolic rate, RER, total motor activity, and sleep time were mostly similar among genotypes (Figure 4F). Total energy expenditure tended to be lower on normal chow and sleep tended to be greater on HFD in female βMUT mice (Figure 4F); however, our study is likely to have been too underpowered to confidently address metabolic changes. In contrast, we were able to detect changes in metabolic indices in global Asic2-deficient mice using similar sample sizes. Although normal circadian differences in light and dark were evident for the variables, there were no differences between genotypes.

### 3.4. Female βMUT Mice Tend to Eat Less Food on Normal Chow

Total food intake was lower in females; however, differences in food intake were abolished on the HFD. The increased variability in food intake in female βMUT mice is not likely due to actual intake but to playing with food (Figure 5A). Time spent eating (Figure 5B), average amount consumed per meal (Figure 5C), and number of uptake events (Figure 5D) were similar among groups. When adjusted for body mass and fat-free mass, total food intake on NFD measured lower in female βMUT mice compared to their WT counterparts, and the difference observed in absolute total food intake on the HFD in the βMUT females was abolished (Figure 5E).

### 3.5. Glucose Tolerance and Plasma Hormones and Lipids Following HFD

Following 5 weeks of the HFD, βMUT mice handled a glucose bolus similarly to WT mice (Figure 6A,B). Fasting plasma insulin, leptin, and ghrelin are shown in Figure 6C, with plasma leptin levels being significantly elevated in male βMUT mice following the HFD. Plasma cholesterol, LDL-C, HDL-C, and triglycerides are shown in Figure 6D and indicate no significant difference between genotypes in both females and males.

### 3.6. βMUT Males Tend to Develop Less Hepatic Steatosis Following a 5-Week HFD

A two-way ANOVA shows total liver mass (Figure 7A) is lower in βMUT male mice. Liver fat content (Figure 7B,C) is reduced and relative lean mass increased in βMUT mice (Figure 7D,E). While liver triglycerides were not different between genotypes (Figure 7F), Oil Red O staining area was significantly reduced in male βMUT mice compared to WT mice (Figure 7G,H). To examine the impact of prevention of steatosis in the liver, we analyzed plasma for liver enzyme indicators of injury ALT, AST, and LDH and found ALT was lower in male βMUT mice (Figure 8).

### 3.7. βMUT Females Exhibit Sex-Dependent Differences in Select Organ Masses and Males Have Larger Abdominal Fat Depots Following a 5-Week HFD

As shown in Table 1, after adjusting for body mass, body length, and tibia length, female βMUT mice had, or tended to have, lower heart and kidney masses. Lastly, while male βMUT mice did not have differences in organ masses, gonadal and visceral fat depot masses were higher when adjusted for body/lean masses and body/tibial lengths despite no differences in absolute and % body fat (Figure 3D,E).

## 4. Discussion

A previous study from our laboratory showed that mice lacking βENaC and Asic2 were protected from diet-induced obesity and hepatic steatosis [13,14]. In follow-up studies, we determined the contribution of individual genes to obesity and hepatic steatosis phenotypes. Because our findings from Asic2 null mice are published elsewhere, the current study investigated the importance of βENaC in the pathogenesis of diet-induced obesity and hepatic steatosis [14]. We first examined hypothalamic expression patterns and metabolic homeostasis in a mouse model carrying loss of function mutations in βENaC. Asic2 is expressed in a greater percentage of hypothalamic neurons important in energy homeostasis, and Asic2 knockout mice show substantial differences in total and basal energy expenditure, motor activity, and food consumption patterns, but no protection from hepatic steatosis [14]. In contrast, hypothalamic βENaC expression is restricted to a much smaller neuron population, and βENaC hypomorph mice show minor changes in metabolic homeostasis yet are prevented from hepatic steatosis.

Given the low number of neurons expressing βENaC, it is not surprising that metabolic homeostasis was minimally altered in βENaC hypomorph mice. Some metabolic homeostasis endpoints can be variable and, coupled with our modest sample size, this likely prevented confident identification of small but significant differences. However, it is unclear what might account for the lower body mass in βENaC hypomorph mice. Despite no measurable differences in energy expenditure or food intake, male βMUT mice exhibited lower body mass compared with WT controls under NFD conditions. Because of little βENaC being expressed in the hypothalamus, it is unlikely that their lower body mass is due to CNS-mediated feeding cues or thermogenesis. Previous studies suggest normal sodium and potassium handling in βMUT mice, discounting the probability of decreased extracellular fluid volume driving the lower body mass [25].

*What is the significance of the shift in fat depots in male mice?* Male βMUT mice display a shift toward greater visceral and gonadal adiposity despite similar total body fat content. Increased visceral fat is associated with inflammation and a higher risk of cardiometabolic disease [26]. Yet, our findings of reduced hepatic steatosis, relative to controls, suggest the opposite. Since fat deposition in the liver originates in part from circulating lipids, we would expect plasma lipids might be lower in male βMUT mice. However, we found circulating HDL-C and LDL-C were similar and triglycerides trended lower in βMUT males (*p* = 0.100, ~15% decrease). The mechanism(s) underlying the shift in lipid distribution is unclear and will be addressed in future studies.

*What is the significance of the reduced heart mass in βENaC hypomorph female mice following diet-induced obesity?* Heart mass was reduced in female, but not male, βMUT mice following a 5-week HFD. This finding was unexpected as βENaC hypomorph mice are modestly hypertensive and we would expect an increase in heart mass as previously reported for NFD-fed mice [19]. The mechanism(s) underlying this outcome are unclear; however, we speculate that this finding might reflect a role for βENaC in cardiomyocyte pressure-induced remodeling. Although not widely regarded, βENaC is expressed in human and murine cardiomyocytes and amiloride-sensitive currents have been reported in cardiomyocytes [26,27,28]. Since ENaC is a member of an evolutionarily conserved family of ion channels that can operate as mechanosensors, we speculate that ENaCs may also play a role in pressure-dependent cardiac remodeling [14,29,30]. A recent report links reduced cardiac fibrosis in a model of diabetes to reduced cardiomyocyte ENaC expression following sodium glucose transporter-2 inhibition, via inhibition of the serum glucocorticoid kinase–ENaC axis [28]. Thus, loss of myocardial βENaC in the hypomorph may disrupt metabolically driven cardiac remodeling. Further studies are needed to understand the metabolic factors and mechanisms underlying this response.

There are limitations associated with this study. First, expression of βENaC protein in hypothalamic regions associated with metabolic homeostasis was not confirmed. Second, the use of a global hypomorph model renders tissue-specific contributions difficult to determine. Future studies using tissue-specific βENaC knockout models will be critical to identifying specific cell and organ involvement mediating alterations in organ and fat depot morphology and sex-specific effects of βENaC function. Second, use of “normal chow” is not the ideal control diet for the HFD, as these diets differ in respects beyond fat content, including carbohydrate and protein sources, fiber types, and the presence of non-nutritive factors such as phytates or phytoestrogens. Not accounting for these various dietary components limits the inferences that can be made in our study.

## 5. Conclusions

The intent of this study was to determine the contribution of βENaC to diet-induced obesity and hepatic steatosis phenotypes observed in mice lacking Asic2 and βENaC. Our findings demonstrate a sex-specific role for βENaC in modulating fat distribution and cardiac remodeling associated with diet-induced obesity, independent of substantial alterations in whole-body energy homeostasis or glucose metabolism. Our results add to the growing body of literature implicating ENaC proteins in physiological mechanisms beyond sodium and water homeostasis. Together, our findings suggest a novel role for βENaC in adipose tissue and cardiac remodeling in the context of diet-induced obesity.

## Figures and Tables

**Figure 1 biology-14-01558-f001:**
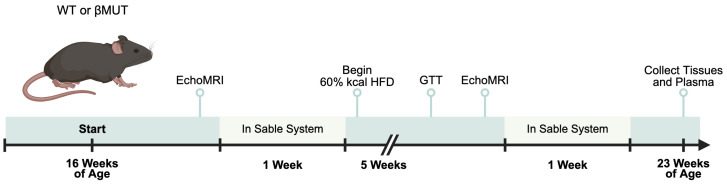
Experimental protocol design. WT—wildtype; βMUT—βENaC hypomorph mouse; HFD—high-fat diet; GTT—glucose tolerance test.

**Figure 2 biology-14-01558-f002:**
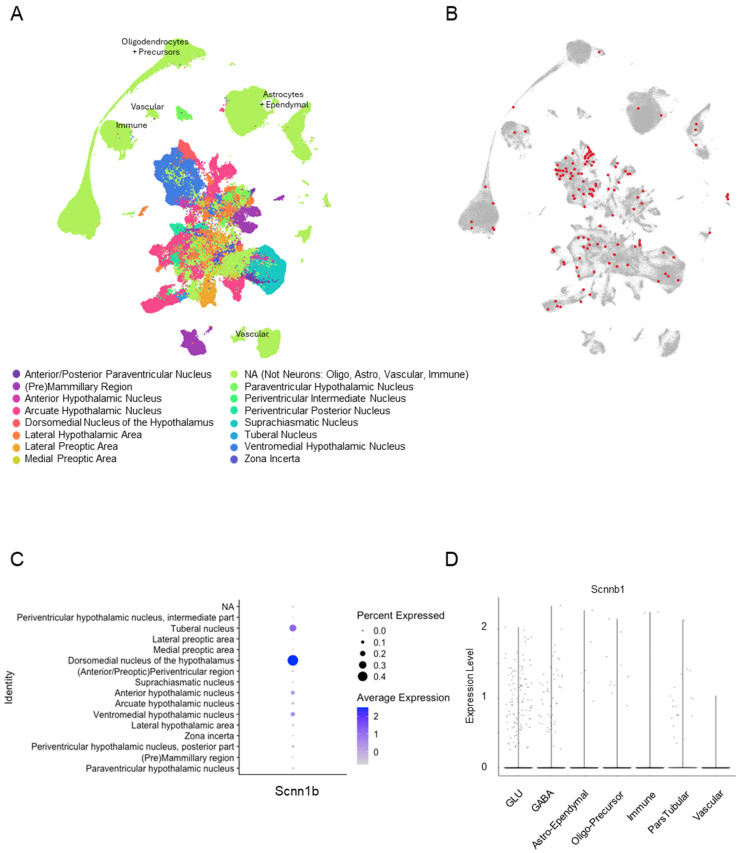
Seurat analysis of HypoMap dataset to determine βENaC subunit expression in the mouse hypothalamus. The HypoMap data set and a user-friendly explorer tool are available at https://cellxgene.cziscience.com/collections (accessed on 24 June 2025). An RDS file was downloaded from the Apollo Repository hosted by Cambridge University (https://www.repository.cam.ac.uk/items/8f9c3683-29fd-44f3-aad5-7acf5e963a75 (accessed on 10 August 2024)). (**A**) A dimensional reduction plot of cluster identification by cell type (“C7_named”). “NA” refers to non-neuronal populations where no marker was present to identify localization, i.e., oligodendrocytes, astrocytes, tanycytes, and ependymal cells. (**B**) Screenshot from the CELLxGENE online explorer tool depicting βENaC subunit expression in hypothalamic regions. Areas of expression (red) generated by authors to enhance the default settings in the online explorer tool. (**C**) Dot plot depicting cluster identity βENaC expression. (**D**) Violin plot showing expression of βENaC in different cell types. From left to right: glutamatergic neurons (GLU), gabaergic neurons (GABA), astrocytes and ependymal cells (Astro-Ependymal), oligodendrocytes and precursor cells (Oligo-Precursor), immune cells, pars tuberalis cells, and vascular cells. Scnn1b—gene for the beta subunit of the epithelial sodium channel. Script: (1A) DimPlot (hypoMap, group.by = “Region_summarized”, label = FALSE). (1C) VlnPlot (object = hypoMap, features = “Scnn1b”, group.by = “Region_summarized”). (1D) VlnPlot (object = hypoMap, features = “Scnn1b”, group.by = “Region_summarized”).

**Figure 3 biology-14-01558-f003:**
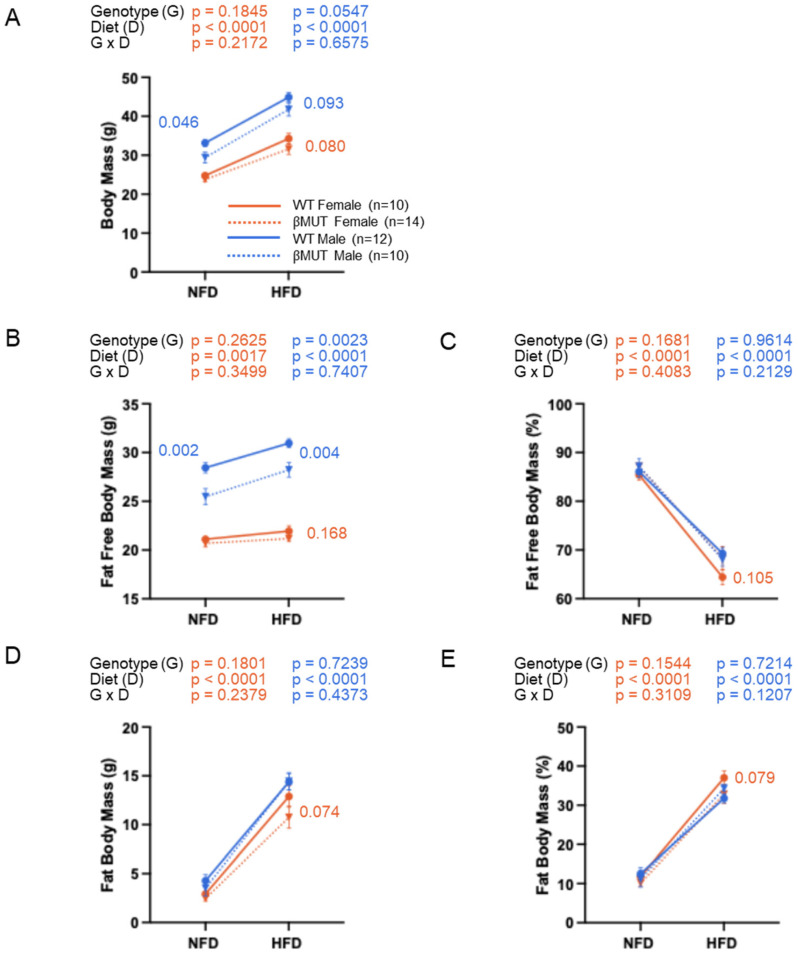
Body composition analysis of WT and βMUT mice. (**A**–**E**) Body mass, absolute and relative fat-free masses, and absolute and relative fat masses before and after a 5-week HFD. Data represent mean ± SEM and were analyzed using a repeated two-way ANOVA for male and female groups, followed by the Fisher Post Hoc test (*p* values for where *p* < 0.200 are shown on panel). WT—wildtype; βMUT—βENaC hypomorph mouse; NFD—normal-fat diet; HFD—high-fat diet.

**Figure 4 biology-14-01558-f004:**
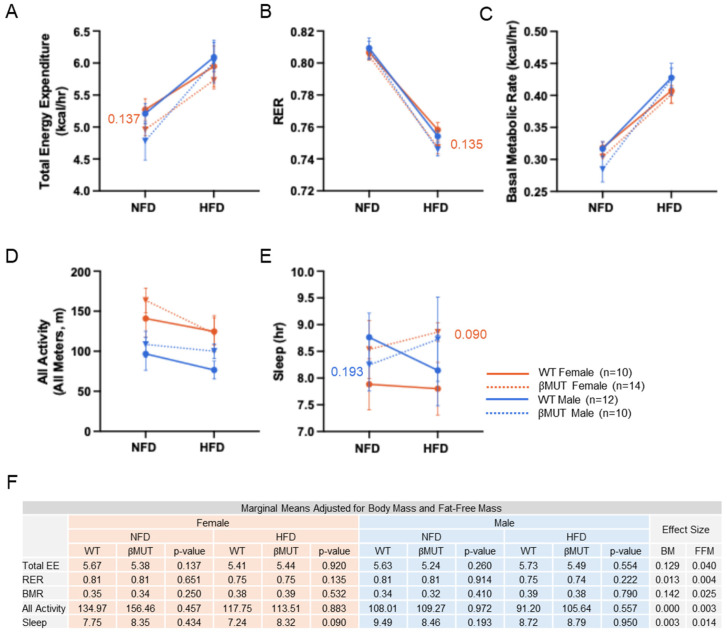
Energy expenditure, RER, BMR, activity, and sleep is minimally altered in βMUT mice on NFD or HFD. (**A**) Total energy expenditure calculated using the Weir equation. (**B**) RER calculated as the ratio of CO_2_ produced to O_2_ consumed. (**C**) Estimated BMR, defined as the mean energy expenditure during the lowest energy expenditure 30 min period. (**D**) All activity quantified as all movement in X, Y, and Z planes. (**E**) Sleep quantified as lack of movement for greater than 40 s. Data for each animal represents the average of three 24-h periods. (**F**) Table of modified means, effect sizes, and *p* values from ANCOVA analysis for each measurement adjusted for fat-free mass and body mass. *p* values less than 0.200 for comparisons of WT and βMUT in each diet condition are shown. WT—wildtype; βMUT—βENaC hypomorph mouse; RER—respiratory exchange ratio; BMR—basal metabolic rate; NFD—normal-fat diet; HFD—high-fat diet.

**Figure 5 biology-14-01558-f005:**
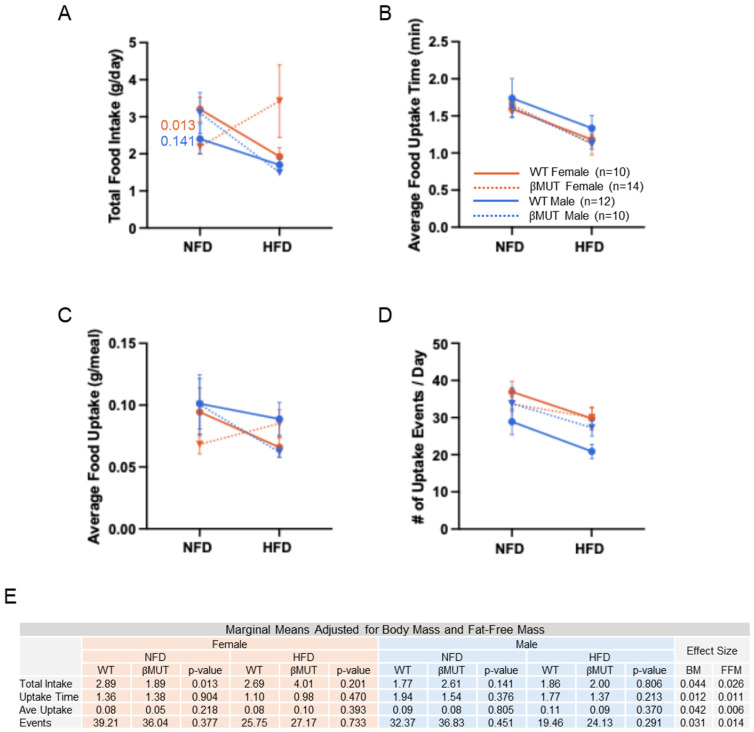
Food intake, intake time, average meal size, and uptake frequency was minimally altered in βMUT mice in NFD and HFD conditions. (**A**) Absolute food intake measured as grams per day. (**B**) Average food uptake time measured in minutes and determined using an internal algorithm. Data for each animal represents the average of three 24 h periods. (**C**) Average Food intake measured as grams per meal. (**D**) Frequency of meals measured as number of uptake events per day. (**E**) Modified means, effect sizes, and *p* values from ANCOVA analysis for each measurement adjusted for fat-free mass and body mass. *p* values less than 0.200 for comparisons of WT and βMUT in each diet condition are shown. WT—wildtype; βMUT—βENaC hypomorph mouse; NFD—normal-fat diet; HFD—high-fat diet.

**Figure 6 biology-14-01558-f006:**
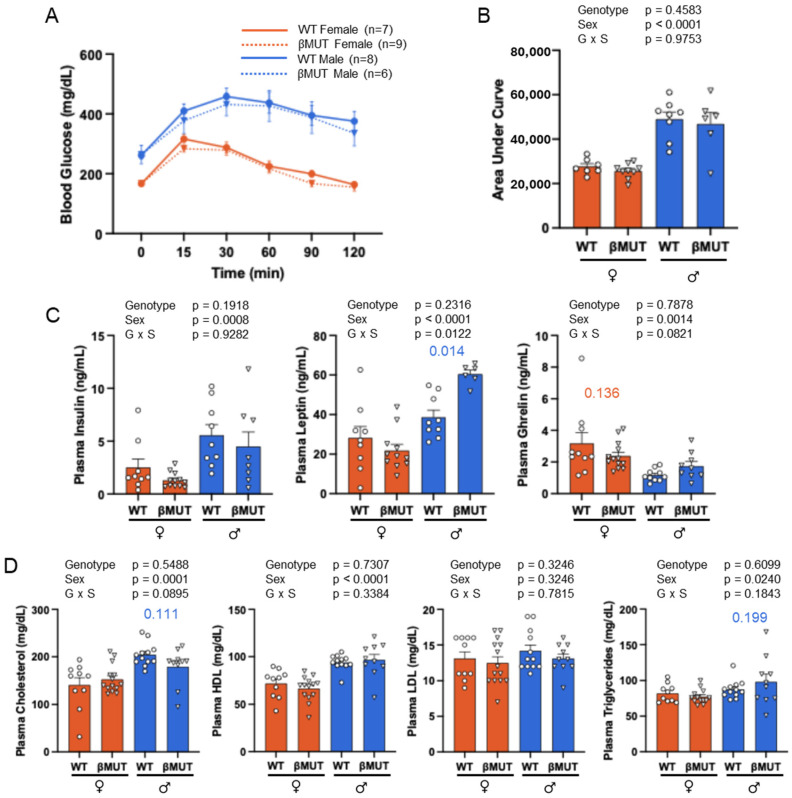
Glucose tolerance and plasma hormone and lipid levels under HFD conditions. (**A**) Glucose tolerance test curves generated by sampling blood glucose at 0, 15, 30, 60, 90, and 120 min post intraperitoneal glucose injection (1 mg/g body weight) (P_genotype_ = 0.4272, RM 3-way ANOVA). (**B**) Total area under the GTT curve was identical between βMUT and WT mice. (**C**) Fasting plasma insulin, leptin, and ghrelin values were determined by ELISA. (**D**) Fasting plasma cholesterol, HDL-C, LDL-C, and triglyceride values were determined by Vet Axel Chemistry Analyzer. *p* values less than 0.200 for comparisons of WT and βMUT in each diet condition are shown. All data represent mean ± SEM. Samples sizes were 6–9 animals/group for glucose and 10–14 for plasma hormones/lipids. Data in (**B**) were analyzed using a two-way ANOVA. WT—wildtype; βMUT—βENaC hypomorph mouse; NFD—normal-fat diet; HFD—high-fat diet.

**Figure 7 biology-14-01558-f007:**
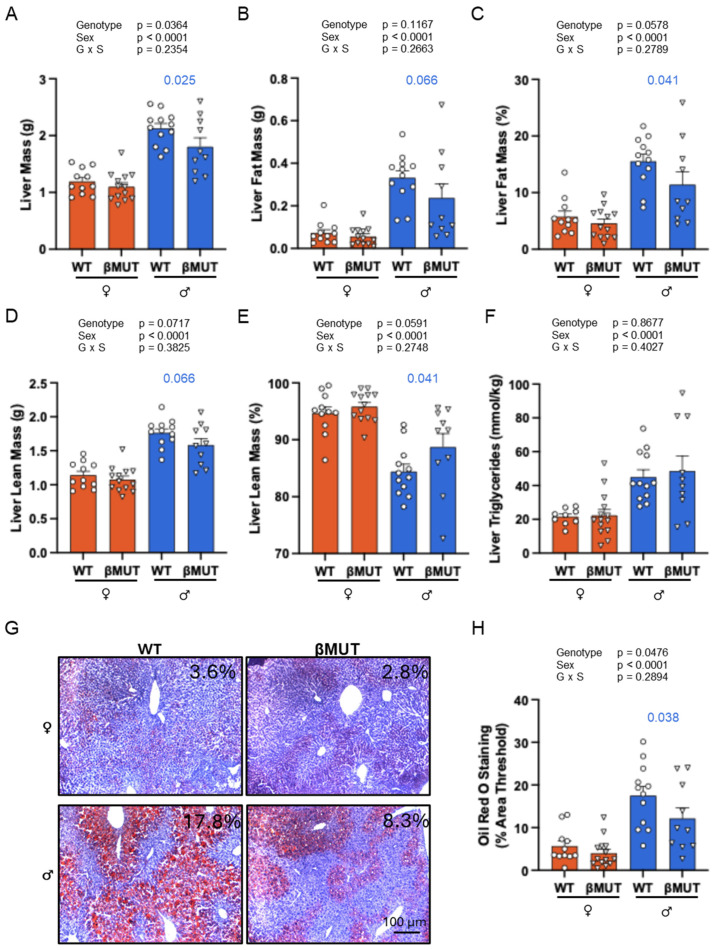
βMUT mice have reduced liver fat accumulation under HFD conditions. Liver fat was determined using multiple approaches. (**A**–**E**) Determination of liver composition using EchoMRI at 15 weeks of age following a 5-week HFD. Liver mass and absolute and relative fat mass was decreased in βMUT male mice (**A**–**C**). Relative lean mass was increased (**E**). (**F**) Liver triglycerides, presented as nmol/µg liver tissue, were not different. (**G**,**H**) Representative images and group data (percent of image area) showing Oil Red O staining was reduced in male βMUT mice. Data represent mean ± SEM, n = 10–14 animals/group, and were analyzed using a two-way ANOVA followed by the Fisher Post Hoc test (*p* values from for where *p* < 0.200 are shown on panel). WT—wildtype; βMUT—βENaC hypomorph mouse; NFD—normal-fat diet; HFD—high-fat diet.

**Figure 8 biology-14-01558-f008:**
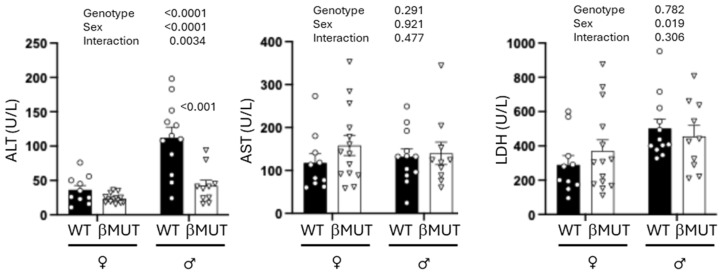
βMUT male mice had reduced plasma levels of liver ALT enzyme level, an indicator of liver injury, following the HFD. Plasma levels of AST and LDH were not different from WT mice. Data presented as mean ± SEM, n = 10–14 animals/group. ALT—alanine aminotransferase; AST—aspartate aminotransferase; LDH—lactate dehydrogenase; WT—wildtype; βMUT—βENaC hypomorph mouse; HFD—high-fat diet.

**Table 1 biology-14-01558-t001:** Marginal Means Adjusted for Body Mass, Body Length, and Tibia Length.

Organ Mass (mg)	Female-HFD			Male-HFD		
	WT	βMUT	*p*-Value	WT	βMUT	*p*-Value
Heart	141	130	0.014	155	157	0.628
Kidney	304	284	0.105	375	388	0.398
Spleen	133	148	0.479	79	80	0.897
Brown Fat	244	147	0.271	152	163	0.67
Gonadal Fat	3441	3095	0.547	1274	1555	0.014
Visceral Fat	2550	1580	0.7	943	1103	0.034
Pancreas	209	199	0.629	205	191	0.491

## Data Availability

Data are available at https://figshare.com/articles/dataset/Data_set_for_Loss_of_bENaC_prevents_hepatic_steatosis_but_promotes_abdominal_fat_deposition_associated_with_a_high-fat_diet_/30460364?file=59100986 (accessed on 16 September 2025).

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
