# Peer review of "Loss of βENaC Prevents Hepatic Steatosis but Promotes Abdominal Fat Deposition Associated with a High-Fat Diet"

_biology, 2025, doi:10.3390/biology14111558_

Round 1

Reviewer 1 Report

Comments and Suggestions for Authors

The authors present a novel role associated with adipose tissue distribution and plausible effect on metabolic homeostasis. In an elegant analysis, they determine how, using a hypomorph mouse model – bMUT; They find differences in fat weight gain, specifically in their anatomical adipose deposition compared to WT mice.

Major observations:

  1. Remove the term "protection against fat deposition in the liver" (lines 15-16). Protection against hepatic fat accumulation involves much more than just controlling a channel protein. It involves modulating the metabolic activity of different hepatocytes in the three zones within hepatocytes. Zone 1 involves fat oxidation and zone 3 involves lipogenesis. In any case, you found that fat accumulation in the liver was "prevented" in that study, but it was not totally protected against, since there are no markers that you analyzed that show this result.
  2. In the section on materials and methods, the authors explain in detail why they decided to use one statistical test or another, justifying when a post hoc test is not applicable due to type I error control (lines 172-174) as well as the distribution of the groups (lines 169-170). Although I understand that statistically it may be convenient, when analyzing the data shown in Figure 6, the points corresponding to each of the mice and their plasma measurements (insulin, leptin, ghrelin, cholesterol, and triglycerides) show a number that does not align with the observation established in the methods. Thus:
    1. Was a normality test performed on each of the measurements?
    2. Or was it performed generally for the male bMUT group?

Because they have one graph with 14 points, another with 9, and others with different numbers.

    1. How many mice are there in total in each group? (Line 27 says 10-14) and ghrelin in male bMUT plasma shows 9.
    2. Justify normality, show p-value of normality test to determine that the correct statistical test was used. With that dispersion between comparisons, it points more to an adjustment for non-parametric tests. Justify.

As one of the strengths of the study, it mention metabolic homeostasis,

    1. Consider a multivariate model, using your plasma measurements, weight measurements, and fat gain of bMUT male mice.
  1. Oil red staining shows clear fat droplet deposition that could suggest macrosteatosis, with mice affected in >20% of hepatocytes.
  2. How many fields per stain per mouse were analyzed? The quantification method and the number of livers (WT n=12 and bMUT n=10) are explained, but not the number of fields analyzed per stain per mouse. I recommend adjusting the image to the value shown in the graph.
  3. The staining of WT mice (Figure 7G WT male red oil staining) appears to be over 70%, whereas the highest mouse reaches 30%, according to the graph in Figure 7H.
  4. It would be appreciated if supplementary material (more representative staining) were attached.

Author Response

Introduction: We thank the reviewers for their thoughtful comments.  We have addressed each comment and modified the manuscript accordingly.  We have also included additional data on plasma markers of liver injury (AST, ALT, LDH) which support of finding that loss of normal bENaC levels in the bMUT models prevents steatosis induced hepatic injury.

Reviewer #1

Summary: “The authors present a novel role associated with adipose tissue distribution and plausible effect on metabolic homeostasis. In an elegant analysis, they determine how, using a hypomorph mouse model – bMUT; They find differences in fat weight gain, specifically in their anatomical adipose deposition compared to WT mice.”

Comment 1: “Remove the term "protection against fat deposition in the liver" (lines 15-16). Protection against hepatic fat accumulation involves much more than just controlling a channel protein. It involves modulating the metabolic activity of different hepatocytes in the three zones within hepatocytes. Zone 1 involves fat oxidation and zone 3 involves lipogenesis. In any case, you found that fat accumulation in the liver was "prevented" in that study, but it was not totally protected against, since there are no markers that you analyzed that show this result.”

Response 1: We have removed the terms “protection” or “protects” and inserted the terms “prevention” or “prevents.” We have also included additional data showing plasma AST (but not ALT or LDH) is substantially lower in male, and trends lower in female, bMUT mice. We have included this data in Figure 8.

Comment 2: “In the section on materials and methods, the authors explain in detail why they decided to use one statistical test or another, justifying when a post hoc test is not applicable due to type I error control (lines 172-174) as well as the distribution of the groups (lines 169-170). Although I understand that statistically it may be convenient, when analyzing the data shown in Figure 6, the points corresponding to each of the mice and their plasma measurements (insulin, leptin, ghrelin, cholesterol, and triglycerides) show a number that does not align with the observation established in the methods. Thus: a) was a normality test performed on each of the measurements or 2) was it performed generally for the male bMUT group?”

Response 2: Thank you for this comment. Our statistician recommends using ANCOVA whenever allometric measurements (i.e. body mass and size) are different among groups.  Thus, we present data as absolute (graphical data) and marginal means adjusted for body mass and fat/lean masses.  We use the statistical analysis from the ANCOVA to identify differences between genotypes. Normality test outcomes are shown in Response 3.

Comment 3: “Because they have one graph with 14 points, another with 9, and others with different numbers. a) How many mice are there in total in each group? (Line 27 says 10-14) and ghrelin in male bMUT plasma shows 9. b) Justify normality, show p-value of normality test to determine that the correct statistical test was used. With that dispersion between comparisons, it points more to an adjustment for non-parametric tests. Justify.”

Response 3: Thank you for this comment. Corresponding group numbers have been updated in each corresponding figure. These studies were performed on an ongoing basis based on availability in our animal colony.  Some groups have fewer animals because we decided to an add assay once the study began or did not have sufficient sample to run all desired tests. We have also performed normality tests using the Shapiro-Wilk test on all data that were determined significant with ANCOVA. P values from the normality tests are provided below:

  • Figure 3A. Male NFD Body Mass. P value SW normality test: WT male=0.186, bMUT male=0.539.
  • Figure 3B. Male NFD Fat Free Body Mass. P value SW normality test: WT male=0.554, bMUT male=0.790.
  • Figure 3B. Male HFD Fat Free Body Mass. P value SW normality test: WT male=0.184, bMUT male=0.650.
  • Figure 5A. Female NFD Total Food Intake. P value SW normality test: WT female=0.600, bMUT female=0.239.
  • Figure 7A. Male Liver Mass. P value SW normality test: WT male=0.957, bMUT male=0.929.
  • Figure 7C. Male Liver Fat Mass %. P value SW normality test: WT male=0.543, bMUT male=0.124.
  • Figure 7E. Male Liver Lean Mass %. P value SW normality test: WT male=0.539, bMUT male=0.072.
  • Figure 7H. Male Oil Red O Staining. P value SW normality test: WT male=0.969, bMUT male=0.881.
  • Table 1. Female Heart Mass. P value SW normality test: WT female=0.051, bMUT female=0.090.
  • Table 1. Male Gonadal Fat Mass. P value SW normality test: WT male=0.478, bMUT male=0.625.
  • Table 1. Male Visceral Fat Mass. P value SW normality test: WT male=0.428, bMUT male=0.310.

Comment 4: “As one of the strengths of the study, it mention metabolic homeostasis, a) consider a multivariate model, using your plasma measurements, weight measurements, and fat gain of bMUT male mice.”

Response 4: Thank you for this comment. The metabolic homeostasis model data were essentially negative. It is unclear if this will provide insight into the mechanism for loss of bENaC in preventing steatosis.

Comment 5: “Oil red staining shows clear fat droplet deposition that could suggest macrosteatosis, with mice affected in >20% of hepatocytes.”

Response 5: We appreciate the reviewer’s insightful observation. While some macrosteatosis may be still present in the beta hypomorph model, the overall reduction in liver mass and percent area of oil red O staining suggests a reduction in hepatic lipid accumulation. Based on Comment 1, we altered the wording in the manuscript to reflect that these mice do not have total protection from hepatic steatosis, but they do have reduced or are prevented from having diet-induced hepatic steatosis to the extent their WT counterparts have.

Comment 6: “How many fields per stain per mouse were analyzed? The quantification method and the number of livers (WT n=12 and bMUT n=10) are explained, but not the number of fields analyzed per stain per mouse. I recommend adjusting the image to the value shown in the graph.”

Response 6: We randomly selected one field from each of two tissue sections per mouse to calculate % thresh-holded area.  Each data point represents the average of the two values per animal (Figure 7H). The methods have been updated to reflect this.

Comment 7: “The staining of WT mice (Figure 7G WT male red oil staining) appears to be over 70%, whereas the highest mouse reaches 30%, according to the graph in Figure 7H.”

Response 7: We use a standard thresholding value for all liver Oil Red O stains performed in our lab. Our histology core uses an automated staining system, thus our Oil Red O signal is standardized from trial to trial. Samples were imaged side by side. The perceived higher area is likely because the threshold is set to capture intensely red cells and avoid light red-pink cells. The threshold values for the representative images shown are as follows WT-M=17.8% , bMut-M-8.3%, WT-F=8.3%, bMut-F=2.8%.

Comment 8: “It would be appreciated if supplementary material (more representative staining) were attached.”

Response 8:  We recognize that this comment likely stems from Comment 7. We have provided new representative images for the WT male and female.  The threshold values for the images are as follows WT-M=17.8% , bMut-M-8.3%, WT-F=8.3%, bMut-F=2.8% and approximate the group mean values.

Reviewer 2 Report

Comments and Suggestions for Authors

An interesting study. However, I have a few suggestions/queries for the authors to address.

The authors have mentioned that “The expression rate of βENaC in the murine hypothalamus is low, with only 0.07% of neurons expressing βENaC mRNA”. 0.07% of the HypoMap dataset size of 384925 cells corresponds to a very small number as you know which may also be due to single cell drop out and integration/batch artefacts. I suggest that the authors perform experiments related to determining the Scnn1b expression in the hypothalamus with RNAscope or IHC.

Line 180: p-values up to “p<0.200” and “no further post-hoc adjustments 173 are necessary”. In my view, these are inappropriate. Please provide the exact p values, 95% confidence intervals and effect sizes for all experimental outcomes. Use appropriate multiple-comparison corrections instead of Fisher LSD. I think it’s better to take the suggestions of a biostatistician for analysis of data.

Please provide detailed methodology on EchoMRI measurements

Lines 289-290: The MS states that there is a decreased Oil Red O area in βMUT males but no difference in biochemical liver triglycerides. I request the authors to repeat the experiment related to measuring triglycerides or determine and confirm the same by HPLC or any colorimetric kit.

Also, TG are reported as nmol/µg tissue. These units are not normally used

Its mentioned that n=10-14 per group. However, several animals were excluded post-hoc. Final sample sizes are not reported clearly in all figures. Please provide the same in the figure legends.

The authors note that normal chow differs from the HFD in many ways apart from fat content. Please discuss at length in what way differences between diets could contribute to the observed changes including organ mass.

Comments on the Quality of English Language

Can be improved by correctiong the grammatical flaws, tense inconsistencies

Author Response

Introduction: We thank the reviewers for their thoughtful comments.  We have addressed each comment and modified the manuscript accordingly.  We have also included additional data on plasma markers of liver injury (AST, ALT, LDH) which support of finding that loss of normal bENaC levels in the bMUT models prevents steatosis induced hepatic injury.

Reviewer #2

Summary: “An interesting study. However, I have a few suggestions/queries for the authors to address.”

Comment 1: “The authors have mentioned that “The expression rate of βENaC in the murine hypothalamus is low, with only 0.07% of neurons expressing βENaC mRNA”. 0.07% of the HypoMap dataset size of 384925 cells corresponds to a very small number as you know which may also be due to single cell drop out and integration/batch artefacts. I suggest that the authors perform experiments related to determining the Scnn1b expression in the hypothalamus with RNAscope or IHC.”

Response 1: We appreciate the reviewer’s thoughtful suggestion to validate Scnn1b expression in the hypothalamus using RNAscope or immunohistochemistry. HypoMap represents a comprehensive, publicly available, data base integrated from 18 different data sets from multiple investigators. We acknowledge the reviewer’s point that single-cell RNA sequencing can be affected by technical limitations such as dropout or batch effects. However, we have recently published data on Asic2 (PMID: 40939131), a closely related molecule, and found very robust distribution and expression.  Thus, the extremely low Scnn1b expression suggests that hypothalamic βENaC expression is likely minimal under baseline conditions, reflecting the point of the figure. We agree that these techniques would provide valuable complementary confirmation of Scnn1b expression at the cellular level in future studies once we can target analysis. Based on the expression levels and our limited expertise in this area, looking for small neuron populations expressing Scnn1b would be akin to “looking for a needle in a haystack”.

Comment 2: “Line 180: p-values up to “p<0.200” and “no further post-hoc adjustments 173 are necessary”. In my view, these are inappropriate. Please provide the exact p values, 95% confidence intervals and effect sizes for all experimental outcomes. Use appropriate multiple-comparison corrections instead of Fisher LSD. I think it’s better to take the suggestions of a biostatistician for analysis of data.”

Response 2: To clarify, we used different statistical approaches for different data sets. For body composition and liver steatosis data sets, we used two-way ANOVA followed by Fisher LSD (with apriori comparisons) and we chose to show p values that some investigators might consider close to significance. Because the allometric data were different among our groups, we used ANCOVA analysis to account for differences in body size in metabolism related and organ masses data sets. The p values from marginal means analysis are provided on the table panel in Figures 4 and 5. Per our statistician, Dr. Lirette, because all estimates come from a single model and are estimated from a pooled post-stratification, the type-I error rate is controlled and no further post-hoc adjustments are necessary for marginal means adjusted for body mass, fat and/or fat free masses. Effect sizes are also provided in the table panels. P values less than 0.200 are provided on the graph to draw the reader’s eye.  We feel strongly that the reader should make the decision if outcomes are “different” with borderline significance values rather than the use of a brightline cutoff for determinations.  In other words, we allow the reader to decide if the differences are meaningful, regardless of if they meet a specific cutoff.  

We removed the allometric analyses for plasma analytes from Figure 6 (and new Figure 8) and have provided the 2-way ANOVA and FLSD post-hoc p values on the graphs.

Comment 3: “Please provide detailed methodology on EchoMRI measurements”

Response 3: We have briefly amended our methodology on EchoMRI. More detailed information based on NMR EchoMRI is available elsewhere.

Comment 4: “Lines 289-290: The MS states that there is a decreased Oil Red O area in βMUT males but no difference in biochemical liver triglycerides. I request the authors to repeat the experiment related to measuring triglycerides or determine and confirm the same by HPLC or any colorimetric kit.”

Response 4: Thank you for this comment. The assay kit can be used for fluorometric or colorimetric quantification, we used fluorometric. We have used this kit in previous investigations and found greater sensitivity in ORO vs triglyceride detection. The discrepancy is likely because ORO stains neutral lipids including triglycerides and cholesterol esters, thus the droplet size may not correspond to triglyceride amount.

Comment 5: “Also, TG are reported as nmol/µg tissue. These units are not normally used”

Response 5: Thank you for this comment. The manuscript has been adjusted accordingly.

Comment 6: “Its mentioned that n=10-14 per group. However, several animals were excluded post-hoc. Final sample sizes are not reported clearly in all figures. Please provide the same in the figure legends.”

Response 6: Thank you for this comment. The sample sizes have been updated in each figure accordingly. Corresponding group numbers have been updated in each corresponding figure. These studies were performed on an ongoing basis based on availability in our animal colony.  Some groups have fewer animals because we decided to an add assay once the study began or did not have sufficient sample to run all desired tests.

Comment 7: “The authors note that normal chow differs from the HFD in many ways apart from fat content. Please discuss at length in what way differences between diets could contribute to the observed changes including organ mass.”

Response 7: We appreciate the reviewer’s comment and agree that the use of a standard chow diet, rather than the purified low-fat control diet matched to the HFD, introduces several variables that could contribute to the observed phenotypic differences. Compositional differences such as energy density and fat source can alter energy intake, nutrient absorption, and gut microbial composition, each of which can influence hepatic lipid metabolism and overall organ mass. Therefore, some of the apparent differences in hepatic lipid content and organ mass between HFD and chow-fed mice likely reflect not only dietary fat content but also broader differences in nutrient composition and metabolic effects inherent to these distinct diet types. We have amended our discussion to reflect this point.

Comment on the Quality of English Language: “Can be improved by correctiong the grammatical flaws, tense inconsistencies”

Response: We have amended the figure legends accordingly.

Round 2

Reviewer 2 Report

Comments and Suggestions for Authors

Most of my queries were answered satisfactorily and necessary changes were made in the MS which is appreciable.

I request the authors to add a sentence in the discussion section acknowledging that βENaC localization has not been validated experimentally (limitation)

The rationale for showing p < 0.200 remains weak. p-values, confidence intervals, and effect sizes may be presented in a supplementary table.

EchoMRI methods: whether scans were performed under fasting or fed conditions?

Regarding the triglyceride units, figures still report values as nmol/µg tissue. Its better to represent in mg/g tissue or mmol/kg

Some figure legends still do not specify n values per group

Author Response

The Reviewer expressed concerns regarding “figures could be improved”.  We want to be responsive to the comments, however, a lack specific concerns provides no direction.  For example, is the font too small? The line point too thin? Are the colors undesirable? We are more than willing to improve the figures, but we are unsure what the Reviewers find problematic. 

Comment 1:  “I request the authors to add a sentence in the discussion section acknowledging that βENaC localization has not been validated experimentally (limitation)”.

Response 1:  We have included a statement in the discussion-limitations stating that bENaC protein expression in hypothalamic neurons associated with metabolic homeostasis have not been validated.

Comment 2: “The rationale for showing p < 0.200 remains weak. p-values, confidence intervals, and effect sizes may be presented in a supplementary table.”

Response 2: We appreciate the Reviewers concern; however, we respectfully disagree with this comment.  In regard to  p values, the American Statistical Association notes the  “p value should not be a substitute for scientific reasoning” on behalf of the reader.  We do not claim those p values are “significant”, we only provide them for the reader to determine if there might be a physiologically relevant difference between groups. As a reader, if I observe what appears to be a difference between groups, but no p value is provided, I wonder how close is the p value to 0.05?  I wonder if the authors use a bright-line cut-off to prevent making a conclusion that didn’t fit their hypothesis. Since we state "metabolic indices on normal chow and HFD were mostly similar between bMUT and WT controls" (line 239), the concern regarding p values and confidence intervals is misplaced (effect sizes are provided in Tables in Fig 4/5).  Additionally, analysis of the metabolic data as covariate analyses (marginal means and effect sizes tables in Fig 4 and 5) is recommended when differences in body size are present (PMID:22205519). 

Comment 3: “EchoMRI methods: whether scans were performed under fasting or fed conditions?”

Response 3: Thank you for the specific question.  The EchoMRI scans were conducted at the same time of day (afternoon) and the mice were not fasted. We have amended our methods accordingly.

Comment 4: “Regarding the triglyceride units, figures still report values as nmol/µg tissue. Its better to represent in mg/g tissue or mmol/kg”

Response 4:  The data were presented as µmol/g in the revised, red-underlined manuscript, which is equivalent to mmol/kg.  We have amended our Y axis title units to reflect this suggestion.

Comment 5:  “Some figure legends still do not specify n values per group.”

Response 5:  The quantitative data (Fig 3-8) are presented with the group sizes on the figure (Fig 3-5) or as individual data points (Fig 6-8). We now provide the group size ranges in the legend for Figs 6-8.